# Deep Learning-Based Weld Contour and Defect Detection from Micrographs of Laser Beam Welded Semi-Finished Products

Christian Nowroth [1,*], Tiansheng Gu [1], Jan Grajczak [2], Sarah Nothdurft [2], Jens Twiefel [1], Jörg Hermsdorf [2], Stefan Kaierle [2,3] and Jörg Wallaschek [1]

1   Institute of Dynamics and Vibration Research, Leibniz University Hannover, An der Universität 1, 30823 Garbsen, Germany; tiansheng.gu@stud.uni-hannover.de (T.G.); twiefel@ids.uni-hannover.de (J.T.); wallaschek@ids.uni-hannover.de (J.W.)
2   Laser Zentrum Hannover e.V., Hollerithallee 8, 30419 Hannover, Germany; j.grajczak@lzh.de (J.G.); s.nothdurft@lzh.de (S.N.); j.hermsdorf@lzh.de (J.H.); s.kaierle@lzh.de (S.K.)
3   Institute of Automation and Transport Technology, Leibniz University Hannover, An der Universität 2, 30823 Garbsen, Germany
*   Correspondence: nowroth@ids.uni-hannover.de; Tel.: +49-511-762-4330

**Abstract:** Laser beam welding is used in many areas of industry and research. There are many strategies and approaches to further improve the weld seam properties in laser beam welding. Metallography is often needed to evaluate welded seams. Typically, the images are examined and evaluated by experts. The evaluation process qualitatively provides the properties of the welds. Particularly in times when artificial intelligence is being used more and more in processes, the quantization of properties that could previously only be determined qualitatively is gaining importance. In this contribution, we propose to use deep learning to perform semantic segmentation of micrographs of complex weld areas to achieve the automatic detection and quantization of weld seam properties. A semantic segmentation dataset is created containing 282 labeled images. The training process is performed with DeepLabv3+. The trained model achieves a value of around 95% for weld contour detection and 76.88% of mean intersection over union (mIoU).

**Keywords:** weld seam; weld defects; deep learning; semantic segmentation; dataset creation; quantization; automatic detection

## 1. Introduction

The weld quality has a direct impact on the performance and lifespan of welded components. Weld defects reduce the weld quality and deteriorate its properties, which is why they should be avoided. There are many approaches to reduce the occurrence of weld defects such as cracks, pores, lack of fusion or incomplete penetration. Therefore, an effective and efficient method of detection and analysis of weld defects is an important topic. To study the physical structure of metals, metallography is typically used for this purpose. After the preparation of the specimens, an expert has to identify the weld seam's properties qualitatively. This evaluation of welds is done manually with the aid of software. The area is marked with lines, whereupon the program specifies it in accordance with the scale. This procedure is time-consuming, and the result depends on the operator. The software cannot reliably distinguish between cracks, pores and other microstructures. It looks for transitions from light to dark areas depending on the set limit, and it can only indicate the roundness of dark particles in bright unetched matrix. For better analysis and comparison between different welds, it is beneficial to describe the weld properties as scalar quantities. However, since the artificial intelligence is being applied more and more in research and industry, quantized values are needed, especially for machine learning. By determining the ratio of significant areas within a micrograph, the effects of parameter changes can be investigated better and compared to each other.

Researchers are gradually applying artificial intelligence to the field of laser welding, such as using the quality inspection system to achieve non-destructive weld measurement and defect detection [1], in-process monitoring based on deep learning [2] or using the semantic segmentation algorithm to detect weld defects in safety vents of power batteries [3]. Long et al. [4] opened the era of fully convolutional networks (FCN) for semantic segmentation. There are currently many variants of FCN-based models that have contributed to the exploration of semantic segmentation [5–9]. Gyasi et al. provide an overview of the use of artificial intelligence in welding technology [10]. Tantrapiwat describes a method of defect detection using a synthetic image dataset if a large number of input images on convolutional neural networks is not accessible [11].

Due to the non-uniformity of the shape, position and size of weld defects, it is a complicated task to manually analyse and evaluate the recorded weld defect patterns. For example, cracks are sometimes wide and sometimes narrow, and there is no fixed standard for length. The pore is not always an ideal circle, and sometimes cracks and pores are connected. These are some of the difficulties in artificially distinguishing the types of weld defects. There are cases in which it is difficult to identify weld defects, such as identifying the boundary of the weld area from the fuzzy heat-affected zone. At the same time, a fair quantitative analysis is also particularly important for the evaluation of welding performance. Therefore, human participation is still always necessary for the analysis of complex weld defects. Hence, it is a very time-consuming and therefore expensive task.

This contribution provides a weld contour and weld defect identification and analysis based on a self-created data set. This leads to convenience for further research in the future. This preliminary work is used to create datasets suitable for processing with machine learning algorithms. From these, predictions can be made about weld quality under different parameter configurations. While many AI-based methods focus on real-time data during the welding process, the deep learning-based detection of structures in the micrographs simplifies the link between real-time data recorded during the welding process and the micrographs evaluated afterwards.

## 2. Methodology

There are many studies on laser weld defect detection. Since different studies require different data sets, a data set of deep welds has been created in order to broaden the application of deep learning in welding processes. It contains micrographs of different welding situations to ensure a wide range of applications in the field of weld property detection. The methodology includes obtaining the micrographs from welds, labeling the obtained micrographs, building the training environment of the neural network and quantifying and analyzing the prediction results.

### 2.1. Data Set Creation and Training Process

Round bars were welded on their circumference either in the form of a bead on plate welds or as dissimilar butt joints. The bars were rotated while a fixed laser beam was used for partial penetration welding. The detailed description of the experimental setup is available in [12]. After welding, the round bars were prepared for metallurgical investigations. Therefore, a cut was made longitudinally with the use of a wet cutting grinder; see Figure 1a. After treatment with an etchant, two high-resolution micrographs were obtained out of one weld; see Figure 1b. To ensure a proper training process, only welds whose weld depth did not reach the center of the round bars were used, since otherwise it is difficult to train the recognition of the outer contour of the weld.

The training process includes image preprocessing, image labeling, neural network training and automatic analysis of prediction results. In this work, the popular model DeepLabv3+ [13] was used, which combines the advantages of multi-scale context information and spatial information, which is very suitable for the task, since the micrographs have weld defects of different shapes and sizes. Some weld defects are very small, which makes identification at high resolutions necessary. DeepLabv3+ combines multi-scale contextual

information and rich spatial information. The model is already packaged in MATLAB [14], and the corresponding backbone pre-trained network can be easily downloaded. All data were split into a training dataset, validation dataset and test dataset using random sampling, but in order to reproduce the results, the random seed was fixed. The training process was carried out with the created dataset using MATLAB version R2020a on a total of four GPUs of model 2080Ti and at least 160 GB of hard disc space reserved. The epochs were set to 50, which turned out to be large enough, because the accuracy of the model hardly changed as the epochs became larger than 20.

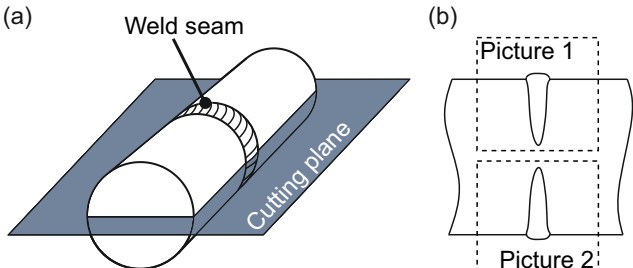

**Figure 1.** Obtaining images from welds: (**a**) Circumferential welds formed during welding of two cylindrical specimens cut as preparation for metallography. (**b**) Two micrographs resulting from metallography.

### 2.2. Preprocessing

Since the size and shape of each class in the micrographs are different and uncertain, in order to allow the neural network to learn the characteristics of each class and distinguish them sufficiently, it is necessary to make the model have a different receptive field. The "atrous spatial pyramid polling" (ASPP) in the DeepLabv3+ model solves this problem. Different dilation rates are used to extract features from images, which gives the model a good understanding of features of different sizes. The use of atrous or dilated convolution ensures that the receptive field is expanded without increasing computational pressure [15]. In the traditional direct upsampling operation, semantic information and spatial information contradict each other. As the number of network layers increases, the feature map will gradually become smaller and the semantic information will become more and more abundant. One pixel covers more information from the original image, but it comes along with the loss of more spatial information. Making a reasonable weight distribution between semantic information and spatial information is very important. In the DeepLabv3+ model, Chen et al. [13] introduced a novel decoder module, which is different from the traditional direct upsampling operation. The low-level feature map is cascaded with the output from the encoder that contains multi-scale rich semantic information.

### 2.3. Image Labeling

At present, the popular image labeling tools for computer vision are "labelme", "labelImg" or "CVAT". In this work, the image labeling tool "Image Labeler" that comes along with the MATLAB software was used. The semantic segmentation model used belongs to supervised learning. In order for the neural network to achieve better results in learning and to clearly identify weld defects and weld metal, each class was labeled. An original image and a labeled image are shown in Figure 2. Both were used as input to the neural network. The original image was needed for the training of the neural network. Finally, the predicted segmented image was compared with the labeled image.

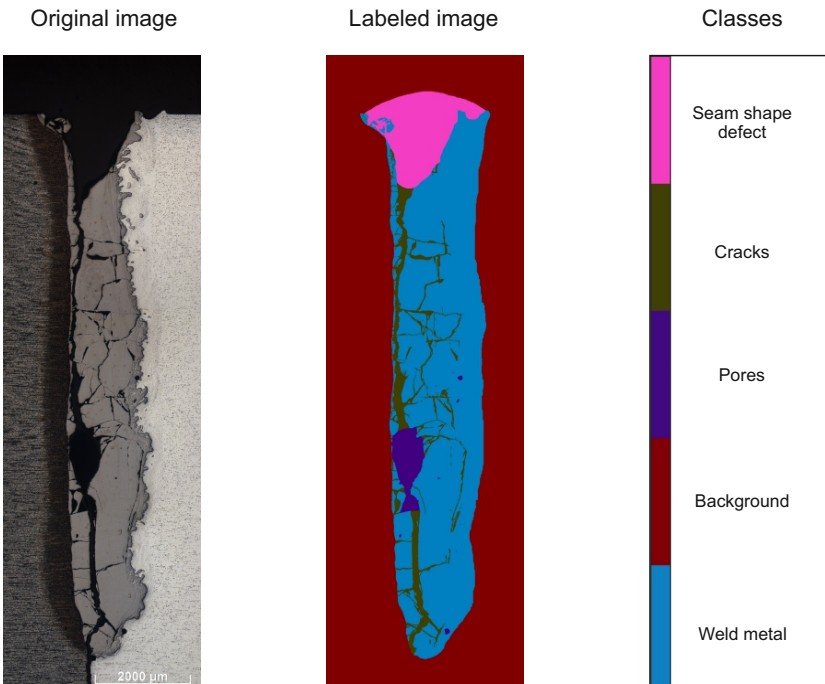

**Figure 2.** Original image, labeled image and class differentiation.

Five classes were initially defined for the classification, as follows:

- The part inside the base material having a clear weld boundary belongs to the weld area, while the rest belongs to the background colored in red;
- A weld reinforcement or sagging that causes a deviation in the expected weld seam shape requirements is defined as a seam shape defect. The label color is pink;
- The remaining parts that are in the weld area without defects are weld metal. The label color of weld metal is blue;
- If there is a long and thin weld defect within the weld area, it is defined as crack. The label color is green;
- Weld defects formed like a bubble are defined as pores. The label color of pores is purple.

Since some defects are very small, image labeling is done pixel-by-pixel. Although there is a clear rule for image labeling, there is still the problem of ambiguous error in the process of image labeling, which can mainly be divided into the following three situations:

- Ambiguous defects between weld metal and background:
  From the original image in Figure 3a, it can be seen that the fusion line of weld metal is difficult to distinguish from the background. The reason is that the colors of weld metal and base metal or heat affected zone are very similar, so no clear boundary can be obtained. In this case, image labeling requires enlarging the image to obtain a finer texture for an artificially based estimation of the position of the boundary;
- Ambiguous defects between pores and cracks:
  The purple label in Figure 3b represents pores. According to the definition, pores are spherical cavities formed by gas inclusions. It should be spherical, but in the actual image labeling, it refers to voids that do not match the thin and long properties of cracks and which are located in the weld area;
- Ambiguous defects between seam shape defects and the background:
  The pink label stands for seam shape defects. Seam shape defects describe the deviation between the actual weld geometry and the expected weld geometry, but it is difficult to determine a fixed shape standard for a weld reinforcement. To solve this problem, in the case that the weld metal is higher than the base metal, it is assumed that there is no seam shape defect. In the case that the weld metal is lower than the

base metal, the rest is artificially filled according to the shape of weld metal, forming a full arc.

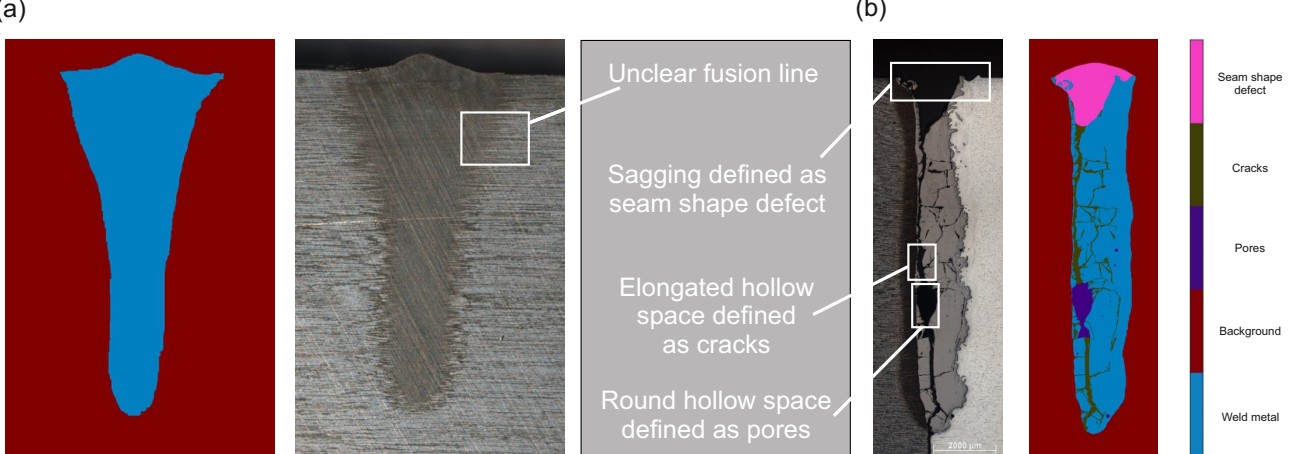

**Figure 3.** Ambiguous defects between weld metal and background (**a**), between porosity and cracks and seam shape defects and background (**b**).

### 2.4. Evaluation Methods

After training the segmentation model, the performance needs to be analysed. Although machine learning technology is advancing, every prediction result has errors. There are some types of evaluation methods that can used to perform a quantitative error consideration of the trained model. A better model can be attained only if the error analysis is continuously combined with the adjustment of the model parameters.

In the field of machine learning and statistical classification problems, the confusion matrix is a visualization tool especially for supervised learning. When evaluating standard machine learning models, the confusion matrix typically is used to divide predictions into four categories: true positives ($TP$), false positives ($FP$), true negatives ($TN$) and false negatives ($FN$). Taking the segmentation result 1 from Section 4.1, Table 1 results as an example of a confusion matrix. With this matrix, it is easy to see if the machine is confusing different classes. Each row of the confusion matrix represents the predicted category and each column represents the true category of the data.

**Table 1.** Confusion matrix example for segmantation result 1; see Section 4.1. $TP$ = blue, $FP$ = yellow, $FN$ = green and $TN$ = red.

|  | Seam Shape Defect | Background | Pores | Cracks | Weld Metal |
|---|---|---|---|---|---|
| Seam shape defect | 661,478 pixels | 7327 pixels | 446 pixels | 13,065 pixels | 170 pixels |
| Background | 4415 pixels | 1,285,278 pixels | 0 pixels | 2 pixels | 1009 pixels |
| Pores | 1793 pixels | 41 pixels | 2318 pixels | 1541 pixels | 94 pixels |
| Cracks | 5413 pixels | 21 pixels | 3011 pixels | 25,964 pixels | 8 pixels |
| Weld metal | 878 pixels | 14,913 pixels | 0 pixels | 155 pixels | 67,812 pixels |

Using these categories, the Intersection over Union (IoU) can be calculated as the overlap rate between the prediction area generated by the model and the area of ground truth. This overlap rate can comprehensively account for the correctly predicted and incorrectly predicted pixels in the prediction results, which fully shows the credibility of the predicted results. It is therefore often used as an important evaluation criterion in the

field of image semantic segmentation. The model can be evaluated using the following equations [16,17]:

$$Accuracy = \frac{TP + TN}{FP + TP + FN + TN} \tag{1}$$

$$Precision = \frac{TP}{FP + TP} \tag{2}$$

$$IoU = \frac{TP}{FP + TP + FN} \tag{3}$$

As there are different methods used to evaluate the model, the results can be seen in Table 2. Equation (1) calculates the ratio of all correctly predicted pixels to all pixels. Equation (2) calculates the probability of being correct among all outcomes predicted for a specific class. Equation (3) calculates the ratio of the intersection of the predicted result and the ground truth to the union of the predicted result and the ground truth for a specific class, but it only considers all cases related to a specific class (TP, FP and FN), and it does not consider the positive effects brought by other classes (TN). As shown in Table 1 with colored cells (TP = blue, FP = yellow, FN = green and TN = red), the use of Equation (3) only considers the results in relation to a particular class. In order to evaluate the welds, the calculation method of Equation (3) is chosen.

**Table 2.** Comparison of different values calculated with the Equations (1)–(3) based on the confusion matrix in Table 1.

| Equation | (1) | (2) | (3) |
|----------|-----|-----|-----|
| Value | 97.41% | 80.96% | 79.74% |

An example is given in Figure 4 as a visual representation. It shows the ground truth and the predicted area as well as the intersection and the union between these two areas.

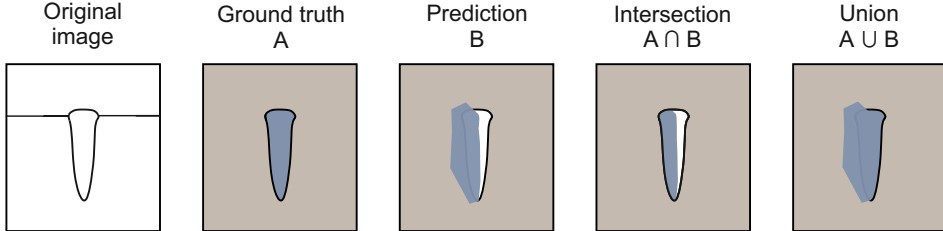

**Figure 4.** Ground truth and prediction area.

As multiple images are considered in general, the IoU of each image in the test set is averaged. Therefore, the "mean IoU" (*mIoU*) is defined as shown in Equation (4).

$$mIoU = \frac{1}{n}(IoU_1 + IoU_2 + \cdots + IoU_n) = \frac{1}{n}\sum_{i=1}^{n} IoU_i \tag{4}$$

To analyse the IoU, the empirical mean and empirical variance are used. The empirical mean is the statistic obtained from one or more random variables, and the empirical variance is an estimate of the variance of the random variable based on the given sample.

### 2.5. Analysis Methods for Prediction Results

With the aid of the trained model, micrographs of laser-beam-welded, semi-finished products are automatically segmented semantically. Each pixel is assigned to a predicted class, which allows the resulting weld to be quantified. Firstly, the weld is analysed using

the number of pixels corresponding to each weld defect. For example, the calculation for the ratio of pores is shown in Equation (5).

$$Ratio_{(Pores)} = \frac{PixelCount_{(Pores)}}{PixelCount_{(Weld\ metal+Cracks+Pores+Seam\ shape\ defect)}} \quad (5)$$

With this calculation, it is possible to compare multiple pictures with different resolutions with each other as the predicted area is related in ratio to the size of the weld area.

## 3. Experimental Results

To evaluate the perfomance of the deep learning model, an ablation study was carried out. Therefore, the influence of different factors on the experimental results, the analysis of the final model and the achievements with the final model are described in detail.

### 3.1. Initialization of Weights

The essence of the deep learning model training process consists of updating the weights. Each parameter must have an appropriate initial value for network training. Poor initialization parameters can cause gradient propagation problems and reduce training speed, while good initialization parameters can speed up convergence and are more likely to find better solutions. In addition, the output of the middle layer of the model is intransparent, and the influence of the previous weights on the output of subsequent neurons is not unique. Even with such advanced parameter passing and updating, an inappropriate initialization of weights can cause laborious parameter learning and even the output loss gradient of the layer activation function to explode or vanish during forward propagation of the deep neural network. In either case, if the loss gradient is too large or too small, it cannot effectively backpropagate, and even if it can backpropagate, it takes longer for the network to reach convergence.

To ensure that low-frequency classes can be learned during the training process, weights were initialized according to the proportion of pixels occupied by each class in all images. The higher the proportion, the lower the initial weight of the class. Artificial adjustments were then made according to the results of the generated images. For example, if the class "cracks" was under-predicted, it was multiplied by a coefficient greater than one in the initial weight. Therefore, many different initial weights were tested. The best combination of initial weights in comparison with the unweighted combination is shown in Table 3. In the case that the backbone was "Xception", the result after adjustment was significantly improved. The initial weights from left to right stand for the weld metal, background, pores, cracks and seam shape defects.

**Table 3.** Comparison of training results with different initial weights.

| Initial Weights | Backbone | Image Resolution | mIoU |
|---|---|---|---|
| 1, 1, 1, 1, 1 | Xception | 2048 × 1024 | 60.96% |
| 0.95, 1.14, 0.15, 0.014, 0.062 | Xception | 2048 × 1024 | 76.88% |

### 3.2. Optimization Algorithm

The function of the optimization algorithm is to minimize or to maximize the loss function by improving the training method. When adjusting the weighting and deviation parameters for model updating, a suitable optimization algorithm can make the model achieve better and faster results. Due to the choice of the DeepLabv3+ model, three optimization algorithms were available: "SGDM", "RMSprop" and "Adam". Details of all optimization algorithms can be found in [18].

To investigate the influence of the pre-trained network on the prediction result, a certain data set of micrographs was used with a variation of the network. From the neural network training results in Table 4, it can be seen that the optimization algorithm of SGDM was more suitable for this project requirements, so SGDM was finally adopted.

**Table 4.** Comparison of training results with different optimization algorithms.

| Optimization Algorithm | Backbone | Image Resolution | mIoU |
|---|---|---|---|
| SGDM | Xception | 2048 × 1024 | 76.88% |
| RMSprop | Xception | 2048 × 1024 | 17.40% |
| Adam | Xception | 2048 × 1024 | 38.08% |

### 3.3. Backbone

The backbone is a pre-trained network used by neural networks for simple feature extraction of the original image. In the Deeplabv3+ model, there are five pre-trained networks available, namely "ResNet-18", "ResNet-50", "MobileNet-v2", "Xception" and "Inception-ResNet-v2". The depth gradually increases from ResNet-18 to Inception-ResNet-v2, and the model needs to learn more parameters. The deeper the pre-trained network, the stronger the learning ability, but therefore the more training data that are needed. For the training process, an image resolution of 2048 × 1024 was used. The neural network was trained with different pre-trained networks with 169, images which is about 60% of the total amount of images of the data set. The results are shown in Table 5.

**Table 5.** Comparison of results with different pre-trained networks with improved parameters.

| Backbone | Image Resolution | mIoU |
|---|---|---|
| ResNet-18 | 2048 × 1024 | 51.14% |
| ResNet-50 | 2048 × 1024 | 71.53% |
| MobileNet-v2 | 2048 × 1024 | 66.15% |
| Xception | 2048 × 1024 | 76.88% |
| Inception-ResNet-v2 | 2048 × 1024 | 57.92% |

### 3.4. Hyperparameter

Hyperparameters include the initial learning rate, the learning rate drop factor, the learning drop period and the regularization coefficient. Adjusting the hyperparameters is computationally intensive and time consuming. With "Xception" as a pre-trained network, two different hyperparameters were used to train the neural network. Specifically, the Nesterov momentum optimizer was parameterized with momentum = 0.9, initial learning rate = 0.05, learning rate drop factor = 0.94, learning drop period = 2 and regularization coefficient = $4 \times 10^{-5}$. The results are shown in Table 6. After comparison, it can be found that hyperparameters have a great impact on the performance of the model.

**Table 6.** Comparison of results with different hyperparameters.

| Hyperparameter | Image Resolution | mIoU |
|---|---|---|
| Original Xception | 2048 × 1024 | 56.49% |
| Improved Xception | 2048 × 1024 | 76.88% |

### 3.5. Amount of Data for Training

Training data were used for the neural network model to learn the properties of each class. The larger the amount of training data, the stronger the ability for the neural network to generalize the properties of each class. An inadequate amount of data can easily lead to overfitting of the neural network because it cannot summarize the rules from more data. To improve the training accuracy, the neural network continuously adapts the model parameters to the characteristics of the training data set, which eventually leads to a high training accuracy, but the test accuracy is not ideal. As the training error decreases, the test error increases instead and overfitting occurs.

To evaluate the influence of the amount of data on the performance of the model, 5%, 10%, 30% and 60% of the total data were used as the training data set. The comparison results are shown in Table 7.

**Table 7.** Comparison of results with different amounts of training data using the pre-trained network "Xception" with improved hyperparameters.

| Amount of Data for Training | Backbone | Image Resolution | mIoU |
|---|---|---|---|
| 14 (5% of the whole images) | Xception | 2048 × 1024 | 37.13% |
| 28 (10% of the whole images) | Xception | 2048 × 1024 | 56.95% |
| 85 (30% of the whole images) | Xception | 2048 × 1024 | 70.12% |
| 169 (60% of the whole images) | Xception | 2048 × 1024 | 76.88% |

Because the amount of 5% is very small, choosing this sample set randomly may lead to different results. To investigate this, the training was conducted with 10 runs in which the sample set was randomly selected from the total data. In this regard, a standard deviation of mIoU of 0.36% has been obtained. Figure 5 shows two original images and the prediction results with different training sets. The larger the training set, the better the prediction results. An increase in the amount of training data improved the overall results. In this case, due to using a pre-trained network and because the change in performance is small in the range from 85 to 169 images, an amount of about 85 images already led to a good compromise between performance and calculation time. Whether a neural network can learn better in a particular detail cannot be answered by the amount of training data, but the more training data that are available, the more statistically significant and stable the training results are.

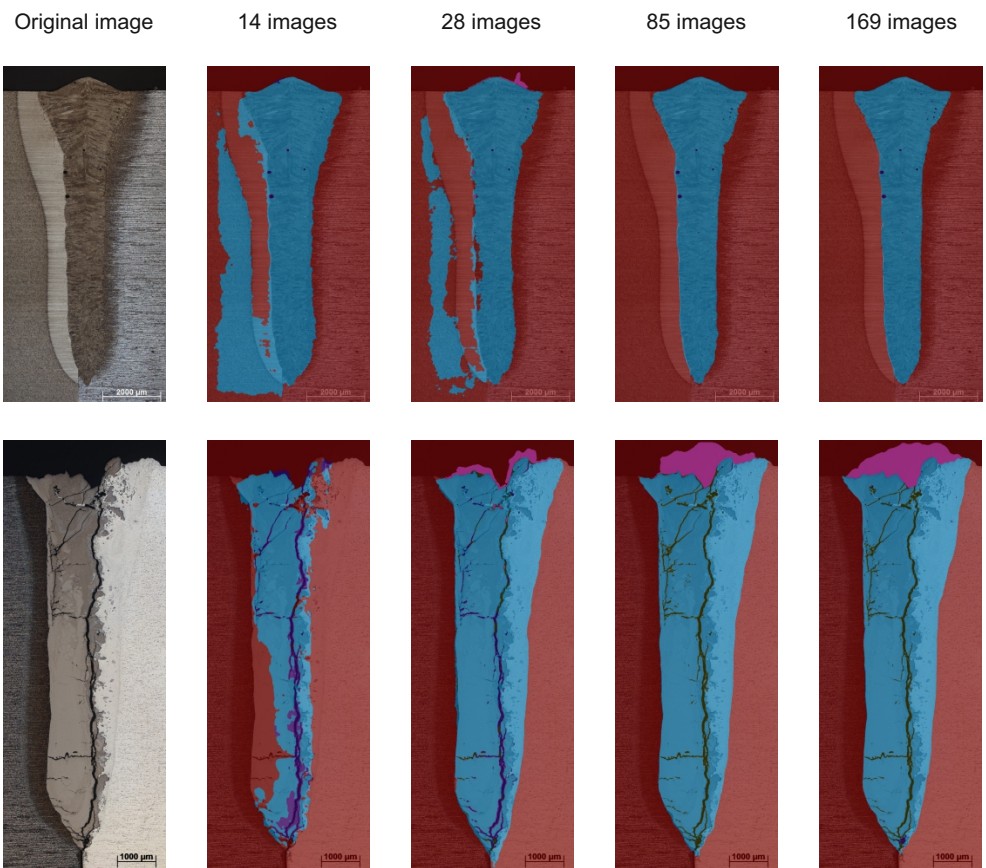

**Figure 5.** Prediction results depending on the amount of training data of 14 to 169 images.

### 3.6. Image Resolution

The image resolution used for neural network training is an important indicator that affects the performance. The micrographs used were high resolution images, and almost every image had a pixel count of 3000 × 2000. If the image is compressed, it is helpful to improve the training speed, but a too low resolution weakens or even destroys some

features; see Figure 6. A crack is a thin and long weld defect. If the resolution of the image is decreased to $512 \times 256$, most of the crack disappears. If the resolution is further changed to $256 \times 128$, almost all of the cracks are not learned. The target is to achieve a good balance of training speed and performance. According to the relationship between the side length of an image and the refinement quality investigated in [19], images with a large resolution are important for the segmentation accuracy. For this reason, the resolution of $2048 \times 1024$ in this case comes along with a sufficient training speed while ensuring that all features are learned as well as possible.

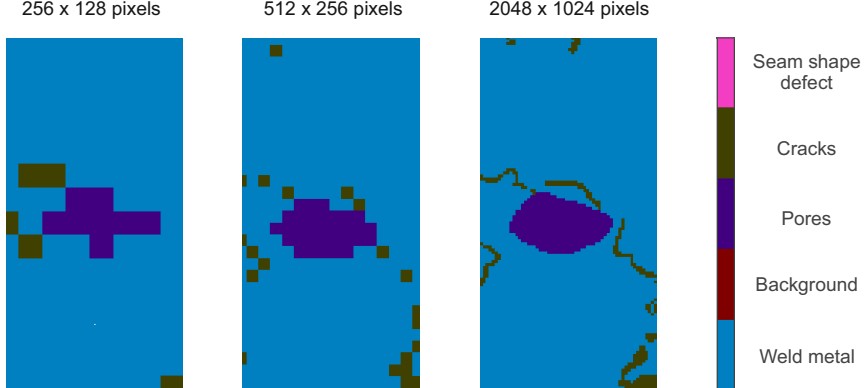

**Figure 6.** Comparison of labeled micrographs with different image resolutions.

## 4. Analysis of the Model

After studying the effect of different factors on the learning ability of neural networks, the DeepLabv3+ model achieved a performance of 76.88% of mIoU on the dataset. According to Equation (3) the following quantitative values for the prediction classes can be obtained; see Tables 8 and 9. The empirical mean and empirical variance of "Weld metal" and "Background" are better than the average, while the other classes are worse. Because of the bigger pixel count of the classes "Weld Metal" and "Background", they are easier to learn in terms of appearance than the other classes.

**Table 8.** Comparison of five classes with image resolution $2048 \times 1024$ based on 57 micrographs.

| Class | Mean Value of IoU | Variance of IoU |
|---|---|---|
| Weld metal | 94.42% | 0.0021 |
| Background | 96.99% | 0.0004 |
| Pores | 65.33% | 0.0905 |
| Cracks | 59.05% | 0.1747 |
| Seam shape defects | 68.59% | 0.1485 |

**Table 9.** Comparison of five classes with image resolution $1024 \times 512$ based on 57 micrographs.

| Class | Mean Value of IoU | Variance of IoU |
|---|---|---|
| Weld metal | 93.47% | 0.0031 |
| Background | 96.82% | 0.0005 |
| Pores | 59.91% | 0.1021 |
| Cracks | 51.43% | 0.1804 |
| Seam shape defects | 72.75% | 0.1431 |

Comparing the two resolutions, it can be seen that a higher resolution leads to a more stable and accurate model. The comparison can be visualized by Figure 7. This decrease in resolution does not seem to have much effect on the overall image, but details are going to be lost. With an enlargement of some areas, some loss of features can be seen, such as some cracks becoming less coherent.

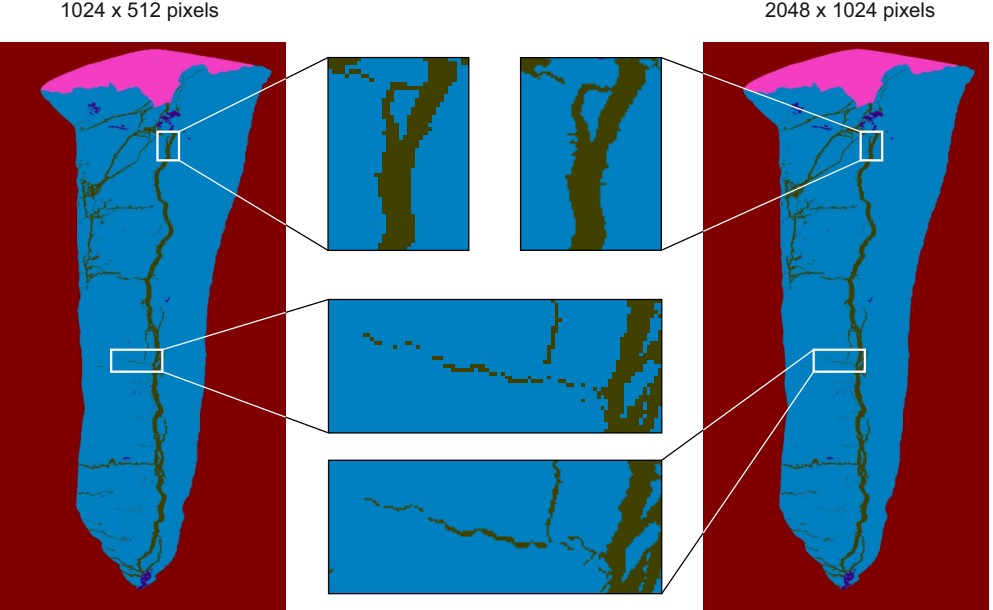

**Figure 7.** Comparison of prediction results with different image resolutions.

## 4.1. Analysis of the Micrographs

With the trained neural network a segmentation result of any micrograph can be parsed as shown in Figure 8. Using the segmentation results, a determination of the pixel counts for each class is possible. The determined pixel counts reflect the severity of the weld defects and according to the IoU algorithm, the reliability of the segmentation results can be seen. The parsed results of Figure 8 are shown in Table 10.

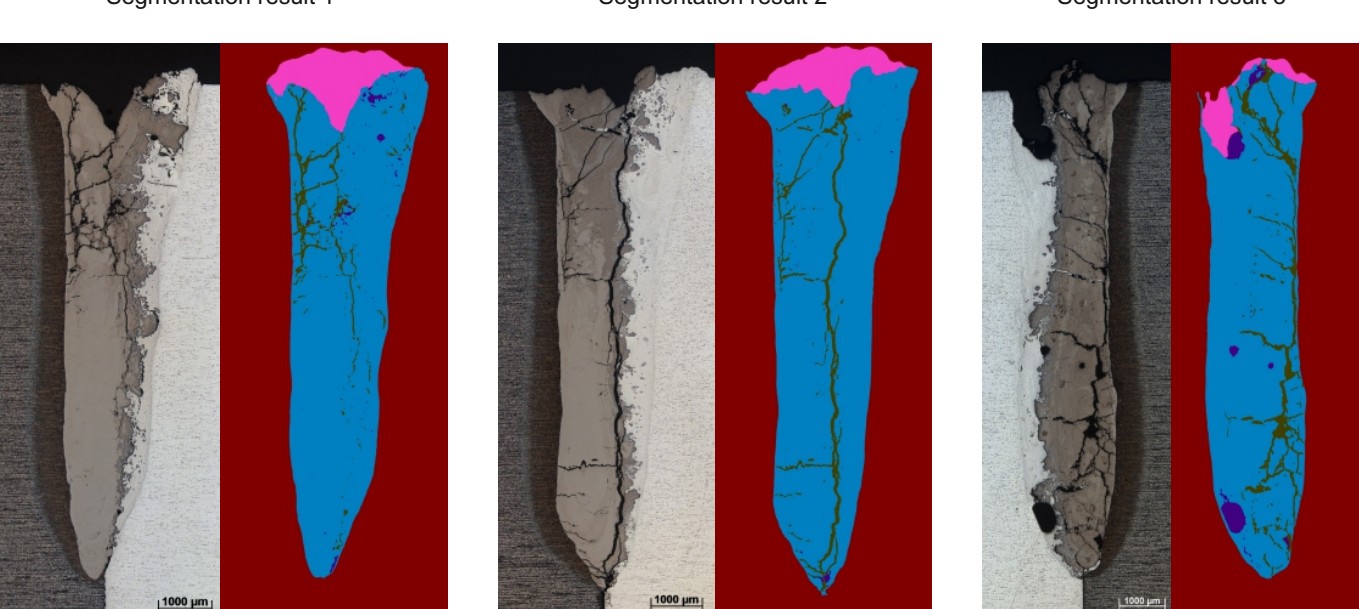

**Figure 8.** Segmentation results.

**Table 10.** Numerical values of segmentation results.

| Segmentation Result | Class | Predicted Pixel Count | Labeled Pixel Count | IoU |
|---|---|---|---|---|
| 1 | Weld metal | $8.6303 \times 10^5$ | $8.5214 \times 10^5$ | 95.17% |
| | Background | $1.6287 \times 10^6$ | $1.6501 \times 10^6$ | 97.89% |
| | Pores | 7329 | 7284 | 25.18% |
| | Cracks | 43,485 | 51,612 | 52.67% |
| | Seam shape defects | $1.0602 \times 10^5$ | 87,489 | 79.76% |
| 2 | Weld metal | $1.0318 \times 10^6$ | $1.0313 \times 10^6$ | 96.20% |
| | Background | $1.5617 \times 10^6$ | $1.5646 \times 10^6$ | 98.13% |
| | Pores | 2033 | 6410 | 15.94% |
| | Cracks | 70,341 | 70,487 | 67.09% |
| | Seam shape defects | 92,548 | 85,578 | 84.28% |
| 3 | Weld metal | $1.0517 \times 10^6$ | $1.0315 \times 10^6$ | 93.85% |
| | Background | $1.8082 \times 10^6$ | $1.84 \times 10^6$ | 97.30% |
| | Pores | 35,269 | 18,796 | 47.90% |
| | Cracks | $1.0546 \times 10^5$ | $1.0574 \times 10^5$ | 65.57% |
| | Seam shape defects | 59,646 | 64,209 | 51.59% |

With the aid of the standardized pixel numbers, the parsed results are comparable despite different scaling of the images. When the certain pixel numbers of defects are known, the ratio of each defect in the weld area can be calculated by Equation (5). According to the data in Table 10, the quantitative values given in Table 11 are calculated. Afterwards, the different parsed results can be compared.

**Table 11.** Comparison of parsed results using the ratio of each defect in the weld area.

| Segmentation Result | Pores | Cracks | Seam Shape Defect |
|---|---|---|---|
| 1 | 0.72% | 4.26% | 10.40% |
| 2 | 0.17% | 5.88% | 7.73% |
| 3 | 2.82% | 8.42% | 4.76% |

## 5. Conclusions

This contribution aims to introduce the current state-of-the-art deep learning techniques to detect weld metal and weld defects. A pre-trained neuronal network was trained and optimized according to the initialization of weights, optimization algorithm, backbone and hyperparameters. By pre-processing the data and adjusting the neural network model, a value of 76.88% of the mIoU was finally obtained for the defined classes. The neural network can automatically detect different areas within micrographs. The detection of weld metal has a high reliability, with an IoU of around 95%. Throughout the research process, a high-definition data set containing 282 images was created. It can be applied to any semantic segmentation model, which has significant implications for future research. Further research can continuously improve the neural network model based on the created data set to achieve better process improvement and quality assurance. With the automated detection of certain features within the weld area, it would also be possible to evaluate the quality of the welds. The results from the deep learning-based weld contour and defect detection could be compared with the established standards, such as DIN EN ISO 13919-1:2020-03, and the weld seam could be evaluated accordingly [20].

**Author Contributions:** Conceptualization, C.N., T.G., J.G., S.N., J.T., J.H., S.K. and J.W.; methodology, C.N., T.G. and J.G.; software, C.N. and T.G.; validation, C.N., T.G. and J.G.; formal analysis, C.N., T.G. and J.G.; investigation, C.N., T.G. and J.G.; data curation, C.N., T.G., J.G. and S.N.; writing-original draft, C.N. and T.G.; writing—review and editing, C.N., T.G., J.G., S.N. J.T. and J.W.; visualization, C.N., T.G. and J.G.; supervision, J.W. and S.K.; project administration, C.N. and J.G.; funding acquisition, J.W. and S.K. All authors have read and agreed to the published version of the manuscript.

**Funding:** Funded by the Deutsche Forschungsgemeinschaft (DFG, German Research Foundation)—CRC 1153, subproject A3—252662854. The authors would like to thank them for the support.

**Institutional Review Board Statement:** Not applicable.

**Informed Consent Statement:** Not applicable.

**Data Availability Statement:** Some or all data generated or used during the study are available from the corresponding author by request.

**Conflicts of Interest:** The authors declare no conflict of interest.

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
