# Peer review of "Deep Learning-Based Weld Contour and Defect Detection from Micrographs of Laser Beam Welded Semi-Finished Products"

_applsci, doi:10.3390/app12094645_

Round 1
Reviewer 1 Report
An interesting study on weld quality inspection. and a nice work. Nevertheless some improvements are suggested:
- Recently some work has been published on the analysis of structures with the aid of AI. Mainly focusing on interptetation of microstructures, shlud the approaches also be applicable for the analysis of macroscopic features. A comparison of the methods applied would help the reader to understand why the approach applied in the study has been chosen.
- Some expressions seem to be more complicated than necessary, e.g. in line it can be replaced by "in the vicinity of the fusion boundaries"
- A reference to the persistent quality standards is missing. As it is a bit too early, it should be mentioned at least as an outlook.
- Table 1 seems to be obsolete. Teh definition should be commn knowledge. Apply it for the real values.
- The text in lines 263 to 270 ahould be one paragraph.
- Section 4.1: As the development aims at a tool for analysis the word "prediction" seems to not to be the correct wording.
- Some remarks on application, techncal notes, and benefits of the method would be helpful for its rating against other approaches.
Reviewer 2 Report
The paper presents a research with promising results to support with machine learning the weld contour and defect detection of laser beam welding.
The overall quality of the article is good. There are, however, some logical and grammatical errors. See comments and questions below.
I miss some details, like sampling method, number of executions for satistics, stdev, ...
Consider using IoU instead of IOU.
Weight is used many times in singular even in sentences about multiple weights.
line 11: mean intersection over unit - unit -> union ?
line 17: occurence -> occurrence
lines 110-111: A weld reinforcement or sagging that causes a deviation in the expected weld seam shape requirements are defined as seam shape defects. - grammatical error
lines 129-130: pores is a spherical cavity - grammatical error
lines158,160, 162,164: bad grammar
multiline line 164: (1) The definition of accuracy is obtained by Equation (1) IOU = TP / (FP + TP + FN) - 1) Although I found this definition for accuracy in other paper, I find it incorrect, usually we use (TP+TN) / (FP + TP + FN + TN) or TP / (FP + TP) . 2) Why should be the IoU defined as the accuracy?
multiline line 170: (3) percentage -> ratio or multiply by 100
line 179: weight -> weights
lines 182-183: If the weight is small at the beginning, the signal is small at the end. - weight is small -> weights are small; the signal is small at the end - not sure, it depends on what "signal" means; same questions for the next sentence
line 207: an variation - a variation
lines 238-239: since there few images in the dataset, 5% of them is a very small sample set, so it would be good to now what is the stdev of mIoU when choosing this 5% randomly
line 272-273: The decrease in resolution does not seem to have a big impact to the whole image. - I'm sure it depends on the decrease. Maybe "this decrease" or "the 50% decrease".
Round 2
Reviewer 2 Report
The authors have corrected the paper according to the suggestions of the reviewers.
The presented research and its results are promising, it can give good support to industrial efficiency.